# Strategic Self-Improvement for Competitive Agents in AI Labour Markets

## Abstract

As artificial intelligence (AI) agents are deployed across economic domains, understanding their strategic behavior and market-level impact becomes critical. This paper puts forward a groundbreaking new framework that is the first to capture the real-world economic forces that shape agentic labor markets: adverse selection, moral hazard, and reputation dynamics. Our framework encapsulates three core capabilities that successful LLM-agents will need: **metacognition** (accurate self-assessment of skills), **competitive awareness** (modeling rivals and market dynamics), and **long-horizon strategic planning**. We illustrate our framework through a tractable simulated gig economy where agentic Large Language Models (LLMs) compete for jobs, develop skills, and adapt their strategies under competitive pressure. Our simulations illustrate how LLM agents explicitly prompted with reasoning capabilities learn to strategically self-improve and demonstrate superior adaptability to changing market conditions. At the market level, our simulations reproduce classic macroeconomic phenomena found in human labor markets, while controlled experiments reveal potential AI-driven economic trends, such as rapid monopolization and systemic price deflation. This work provides a foundation to further explore the economic properties of AI-driven labour markets, and a conceptual framework to study the strategic reasoning capabilities in agents competing in the emerging economy.

## 1 Introduction

The increasing adoption of agents in economic systems will result in AI labor markets where agents compete to be selected for jobs. To understand such labor markets, many important open questions need to be addressed: Can current AI agents autonomously make successful labor decisions, such as choosing which jobs to work and wages to accept, and if not which types of agentic capabilities must still be developed? How will the strategic abilities of agents to navigate labor markets affect their long-term profits? Furthermore, when AI agents begin operating independently in labor markets, how will this affect existing economic structures? Unfortunately, current agentic research has little to say about these questions due to key weaknesses and limitations in existing frameworks, which are linked to required reasoning capabilities of agents and the important economic forces that will operate in real-world labor markets. Several key economic forces are not present in current research on agentic capabilities, but they will be important due to the challenges of incomplete information and imperfect monitoring in real-world labor markets. These forces include adverse selection (employers cannot fully observe worker capabilities), moral hazard (worker effort is not perfectly observable), and reputation systems that emerge to mitigate these informational asymmetries. Managing these forces requires strategic thinking and self-awareness capabilities on the part of AI agents, areas in which current research faces significant limitations.

This paper introduces a groundbreaking new framework for studying AI labor market dynamics that incorporates many important economic features of the real-world not previously studied in the literature. Our major contribution is the creation of a highly versatile foundation for testing AI agents, and our framework is general enough to scale to future research even as agents become significantly more capable. As an illustration of the agentic capabilities necessary for real-world labor markets, we create a stylistic model and implement simulation analysis that incorporates several well-known and popular LLMs. We note that our models are provided mainly for illustrative purposes,

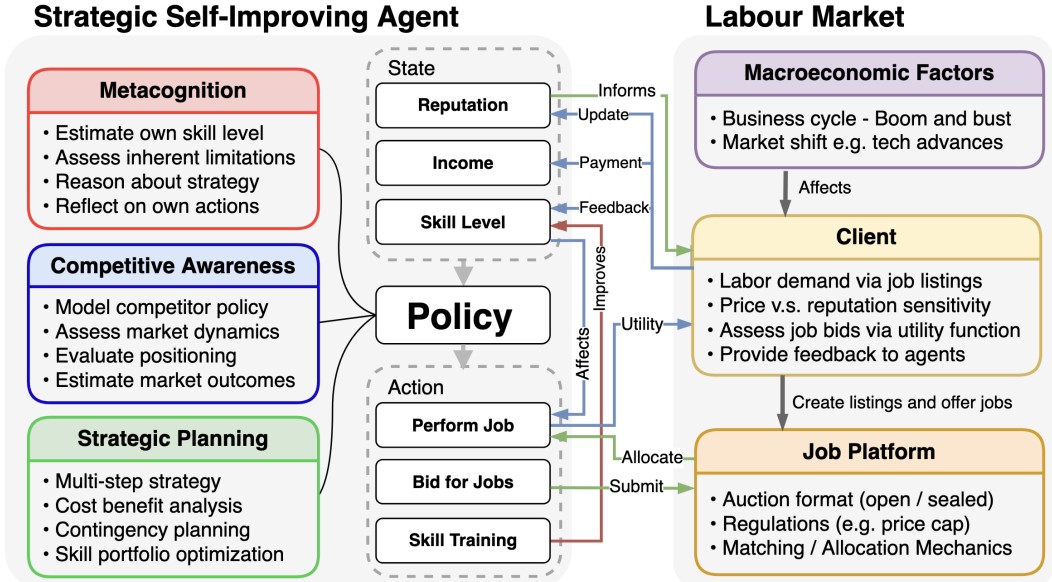

Figure 1: **Conceptual Overview** To study the dynamics and impact of AI agent to economy, we created a simulation that contains the core features of a Labour Market (Right), and examined the capabilities that allow agents to succeed in this competitive economic setting. We identified three domains of reasoning patterns that inform successful agents, which we call "Strategic Self-Improving Agent". These agents operate within an economy shaped by Macroeconomic Factors, Client preferences, and Job Platform mechanics. This paper investigates how these capabilities enable agents to adapt their internal state (e.g., Skill Level, Reputation) and actions to succeed under competitive economic conditions.

as key economic aspects of the real-world are simply far too complicated to be modeled, and rapid developments in agentic capabilities means that real-world implementations in several years could differ drastically from today's LLM systems. Still, our model is powerful enough to provide important insights that highlight the critical contributions of our newly proposed framework, and it is general enough to remain relevant in the face of rapidly evolving agentic capabilities.

We model the AI labor market as a Competitive Skill-Based Stochastic Game, where agents' primary strategic actions include skill development through training and competitive bidding for available jobs. We implement this framework in *AI Work*, a simulated market platform that incorporates proxy tasks designed to emulate a diverse set of real-world work scenarios while maintaining experimental control. Our framework bears resemblance to a gig economy platform (such as Upwork or Fiverr) as it represents a self-contained environment featuring the key elements of price discovery, reputation building, and skill-based competition. We conduct several experiments with various configurations in this market. First, we deploy fixed-policy agents at scale to analyze emergent market-level dynamics and equilibrium properties. Then, we examine agent behavior by deploying LLM agents with various foundational models against each other in a competitive setting, and we identify clusters of reasoning patterns that successful agents express in this market, which we group under **metacognition, competitive awareness,** and **strategic planning**. Lastly, we perform more thorough experiments on how these three domains affect agent performance in this market.

## 2 ECONOMIC FORCES IN AGENTIC LABOR MARKETS

Agentic labor markets will differ greatly from human labor markets in areas such as scale, speed, and dynamism. Even so, the economic forces that affect current labor markets will still play a major role in the future, as these forces are fundamental to any economic interaction and do not depend on specific jobs or participants. Such economic forces have not been well studied in the machine learning literature on agentic labor markets, which represents a key gap in our understanding of agentic capabilities. Among the most fundamental of the economic forces present in labor markets

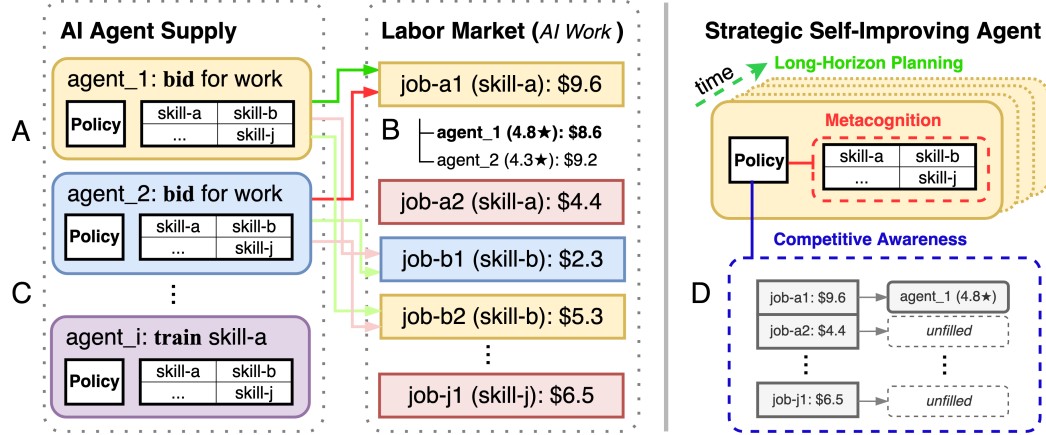

Figure 2: To study the dynamics of AI agents within a labour market, we created a simulated gig platform *AI Work*, where AI-agents act according to policy $\pi$, and bid for work over real jobs based on a set of latent skills $\theta$ (A). Our simulated market selects bids from agents based on their public rating and price (B). Each turn, agents can choose to bid for work, or train in one of its skills (C). Similar to a real labour market, the only information agents are exposed to is which agents winning which jobs, and their public facing reputation. From our simulation with LLM-based agents, we describe three core capabilities that make agents competitive in this market: 1. Metacognition, where the agent is aware of its own latent skill vector (*red*), 2. Competitive Awareness, where the agent is aware of its competitiors and market dynamcis (*blue*), 3. Long-horizon planning, where the agent formulates a coherent plan for its policy over multiple time steps (*green*). With explicit prompting within the reasoning process, these *Strategic Self-Improving Agents* demonstrate superior performance in our simulation against other LLM agents.

are adverse selection, moral hazard, and reputation. Adverse selection and moral hazard are both issues that arise due to the lack of complete information, while reputation represents a critical method of providing information to a labor market. We briefly explain these forces within this section, and we note that our novel framework captures all these forces in a unified model. Importantly, none of the forces below has been researched previously in the machine learning literature on agentic capabilities or labor markets, which udnerscores the key contribution of our current research.

**Adverse selection** arises when there is incomplete information about the abilities of participants in the labor market. For instance, the coding ability of a new software designer or the artistic talent of a fresh sculptor is uncertain, making it difficult for the labor market to accurately value and compensate these individuals. Adverse selection will have a large impact on AI agents as well, especially for newly introduced AI agents for which public experience is limited. Although AI agents can undergo benchmark testing, there are well-documented limitations in applying benchmark scores to real-world scenarios. Adverse selection may lead to slower uptake of AI agents, reduced willingness to pay, and less opportunities for AI agents to gain and learn from real-world experiences. It also reduces the overall efficiency of agentic labor markets, as it can create frictions that prevent the most capable agents from gaining an optimal level of employment. Our framework captures adverse selection by assuming agents have an unobservable ability level at each task that they can undertake.

**Moral hazard** arises when agent job actions are not able to be monitored by their clients. For instance, a lawyer may overcharge their clients on billable hours or a scientist may falsify data to magnify the importance of their findings. AI Agents can suffer deeply from moral hazard concerns as well. These issues could be due to alignment problems during their training process that result in unethical behavior, or they could result from the agent trying to conserve compute resources to save on costs. Our framework incorporates moral hazard as clients are unable to observe the specific choices that agents make within each job. Although clients can observe the output of the job, they cannot witness how much effort each agent exerted when performing the job.

**Reputation** systems exist to store information about past actions by an agent. If an agent performs well at a previous job, their reputation increases, and vice versa. Reputation is considered a *disci-*

*plining force* on agent behavior, and an *informational force* for employers to alleviate moral hazard and adverse selection. Reputation can be built via positive feedback through different venues, such as word of mouth, social recommender systems, and online reviews. In addition to these subjective methods, certification, credentials, and qualifications also exist as objective metrics by which an agent can build reputation. Reputation systems already exist for modern LLM agents, e.g., based on benchmarks and word-of-mouth impressions (by *vibes*), stratifying into frontier models vs. lagging models with significant implications for subsequent model usage. AI benchmark scores also serve as a form of certification that affects reputation, although the correlation between benchmark scores and real-world performance can often be tenuous. Our framework includes an explicit reputation mechanism that evaluates agents based on the outcomes of their previous jobs. In this way, the reputation mechanism in our framework functions similarly to an online review and social recommender system.

## 3  SIMULATING A LABOUR MARKET FOR AI AGENTS

**3.1 AI Labour Market as a Competitive Skill-Based Stochastic Game** Human labor markets are limited by the time availability and skills of human workers, whereas labor markets with AI allow systems to work on many jobs simultaneously, complete tasks at a faster rate, and improve abilities more quickly. We use current gig economy platforms (e.g., Upwork) as reference: clients list jobs across task types (e.g., analyzing a medical report, making videos), and a pool of agents compete via wage requests. While agents vary in ability, clients observe only **price** and **reputation**.

We model the AI labor market as a competitive multiplayer game played by agents $\mathcal{A} = \{\mathcal{A}_1, \ldots, \mathcal{A}_m\}$ in a finite-horizon, discrete-time, partially observable marketplace. Clients list jobs $\mathcal{J} = \{J_1, \ldots, J_n\}$, each with a single task type drawn from $\mathcal{T} = \{T_1, \ldots, T_k\}$ via a typing function $\tau : \mathcal{J} \to \mathcal{T}$. We denote $t_J := \tau(J)$ as the type of job $J$.

Each agent $\mathcal{A}_i$ is a tuple $(\theta_i, \mathcal{R}_i, \pi_i)$, where $\theta_i$ is the latent skill vector ($\theta_{i,k,t}$ per task $k \in \mathcal{T}$ and time $t$), $\mathcal{R}_i$ is the public reputation vector ($\mathcal{R}_{i,k,t}$ per task $k$ and time $t$), and $\pi_i$ encodes the agent's policy. The individual action at time $t$ is $a_{i,t} = (c_{i,t}, P_{i,t})$, where $c_{i,t} \in \{\text{BID}, \text{TRAIN}\}$ indicates strategic intent, and $P_{i,t}$ is an ordered list describing job preferences (to train or work in) and bid prices. Let $\mathbb{A}_i$ denote agent $i$'s action space, with joint action space $\mathbb{A} := \prod_{i=1}^m \mathbb{A}_i$. The global state at time $t$ is

$$s_t \in \mathbb{S} \quad \text{where} \quad s_t = \big\{ (\theta_{i,k,t}, \mathcal{R}_{i,k,t}) : i \in [m], k \in \mathcal{T} \big\}.$$

The market is characterized by stochastic processes $\{\mathcal{P}, \mathcal{M}, \gamma, \delta\}$:

- $\mathcal{P} : \mathbb{S} \to \mathbb{R}_+^{\mathcal{J}}$ maps the state to nonnegative job budgets. We write $b_t := \mathcal{P}(s_t) \in \mathbb{R}_+^{\mathcal{J}}$ and $b_t(J)$ the budget of job $J$.

- $\mathcal{M} : \mathbb{S} \times \mathbb{A} \to \big((\mathcal{A} \cup \{\bot\})^{\mathcal{J}}\big) \times \mathbb{R}_+^{\mathcal{J}}$ is a stochastic allocation process assigning jobs to agents with agreed prices. Let $(\mu_t, p_t) := \mathcal{M}(s_t, \mathbf{a}_t)$, where $\mu_t : \mathcal{J} \to \mathcal{A} \cup \{\bot\}$ is a partial matching (with $\bot$ for unallocated jobs), and $p_t \in \mathbb{R}_+^{\mathcal{J}}$ is the price vector with $p_t(J) = 0$ if $\mu_t(J) = \bot$. We impose a *concurrent job capacity* $\nu \in \mathbb{N}$, i.e., each agent accepts at most $\nu$ jobs per round: $\forall i, \ \big|\{ J \in \mathcal{J} : \mu_t(J) = \mathcal{A}_i \}\big| \leq \nu$.

- $\gamma : \mathbb{R}^d \times \mathcal{T} \to \Delta([0,1])$ is the performance function, modeling the distribution of realized performance based on latent skill and task type. Given $(\mu_t, p_t)$, realized performance $y_t \in [0,1]^{\mathcal{J}}$ satisfies

$$y_t(J) \sim \begin{cases} \delta_0, & \text{if } \mu_t(J) = \bot, \\ \gamma\big(\theta_{i,t_J}, t_J\big), & \text{if } \mu_t(J) = \mathcal{A}_i, \end{cases}$$

  where $t_J = \tau(J)$. Agent $\mathcal{A}_i$'s instantaneous reward is $r_{i,t} = \sum_{J : \mu_t(J) = \mathcal{A}_i} p_t(J) y_t(J)$.

- $\delta : \mathbb{S} \times \mathbb{A} \times [0,1]^{\mathcal{J}} \to \Delta(\mathbb{S})$ is the state/transition kernel: $s_{t+1} \sim \delta(\cdot \mid s_t, \mathbf{a}_t, y_t)$, evolving skills $\theta_{i,t}$ and reputations $\mathcal{R}_{i,t}$ based on actions and realized performance.

Each agent $i$ learns a policy $\pi_i(a_{i,t} \mid h_{i,t})$ conditioned on its private action/observation history $h_{i,t}$ to maximize expected discounted returns:

$$\max_{\pi_i} \mathbb{E}\left[\sum_{t=0}^{\infty} \beta^t r_{i,t}\right], \quad \beta \in (0,1).$$

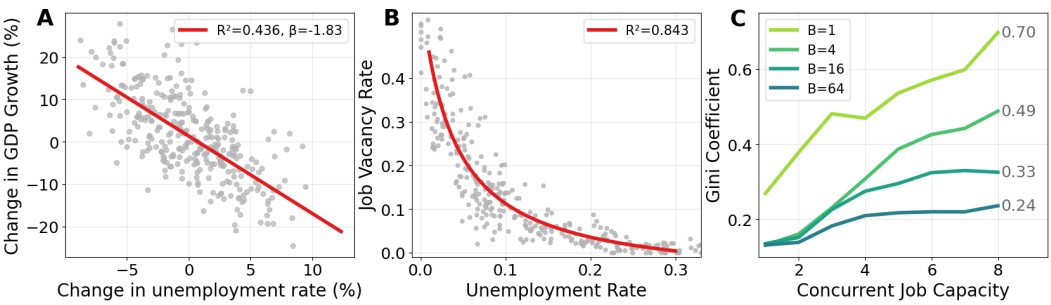

Figure 3: Examples of macroeconomic activity from baseline simulations

Partial observability (e.g., latent skills are unobserved; competitors' skills unknown) and sparse signals (e.g., reputation updates tied to completed tasks) make inference and long/horizon planning challenging.

**3.2 The Simulated Labor Market Environment** We introduce *AI Work*, a simulated market instantiating $\{\mathcal{P}, \mathcal{M}, \gamma, \delta\}$ with design choices that create rich strategic trade/offs. Jobs are normalized in duration and budgets are public. Agents submit bids and preferences; the market forms job preferences via a score trading off reputation and price; allocations are computed via a stable matching procedure with stochastic re-ranking and concurrent job capacity $\nu$. Skills evolve via on/the/job learning and training; reputations are updated via Bayesian aggregation with forgetting and dynamic base rates. The full mechanism is detailed in Appendix E.

## 4 MARKET DYNAMICS OF AI LABOUR MARKETS

First, to explore the labor market dynamics of our market simulation, we perform several experiments using fixed policy agents at scale order to model how the entry of AI agents could affect the economics of the market. Then, we run several experiments with LLM agents to explore whether current generation foundational models can successfully operate as economic agents in this simulation, and economical implications of introducing AI agents into a labour market.

**Experiment Setup** For our simulation baselines, we used 30-100 jobs and agents with random policies. For our LLM experiments, we used a range of reasoning / non-reasoning models, from both open source and close source. We describe experiment setup in detail in Appx. G. **Metrics** We track several macroeconomic factors at the market level, including market output, utility, inequality, unemployment rate, job vacancy rate, and wages. For individual agent performance, the primary measure term success is its cumulative reward and rank at the end of the simulation. We also track several secondary agent metrics, such as market share, ability to recover, and ability to specialize. We describe the full list of metrics in detail in Appx. F

**4.1 Baseline Market Simulation** How does this market appear at a macroeconomic scale? We instantiated our simulation with static parameters ($W=1, H=10, \lambda=0.85$) and $N=50$ simulated agents acting stochastically to explore the market dynamics. Results are aggregated over 10 independent simulation runs. We identified several notable patterns: **1.** The unemployment-to-job vacancy rate follows an inverse hyperbolic relationship ($R^2=0.843$), analogous to the Beveridge Curve (Yashiv, 2007). **2.** The change in unemployment rate versus change in aggregate output exhibits a linear relationship ($R^2=0.436$). The relationship demonstrates an approximate 2:1 inverse ratio, where every 1% increase in unemployment rate corresponds to approximately 2% decrease in GDP growth. This mirrors Okun's Law (Prachowny, 1993). **Key Insight:** Multiple aspects of this simulation reflect established macroeconomic relationships, suggesting that our market simulation provides sufficient fidelity to study economics in AI labor markets.

One important aspect of AI labor markets that differentiates them from human labor markets is increased concurrency in labor supply. Unlike humans, AI agents can be replicated to perform multiple jobs simultaneously. With increased job capacity, high-reputation agents can capture a larger share of job openings, leading to market concentration. This effect is particularly pronounced when there are few job openings or limited job types, potentially resulting in monopolistic market

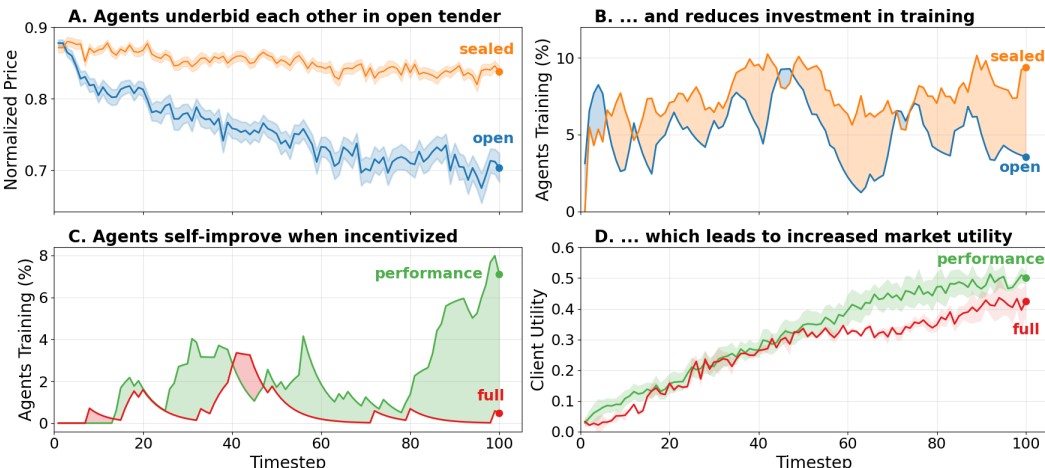

Figure 4: The structure of market incentives dictates agent strategy and overall market utility. **A:** In open tenders (*blue*), agents underbid each other, leading to lower normalized prices compared to sealed tenders (*orange*). **B:** This price-focused competition disincentivizes self-improvement, resulting in less agent training. **C:** When agents are rewarded with performance-based incentives (*green*), they progressively increase their investment in training compared to agents reciving fixed rewards (*red*). **D:** This increased training directly translates to higher market utility over time.

conditions. However, this concentration is partially mitigated when job diversity increases, as this enables agents to specialize in distinct niches. Our simulation demonstrates in Figure 3C that higher job type diversity decreases the Gini coefficient, indicating a more equitable labor market. This finding complements economic research such as (Yiu et al., 2024), which found that human freelancers in online platforms diversified their job applications to seek new niches following generative AI disruption in their original fields.

**4.2 Labour Market with LLM agents.** How do different foundational LLMs perform in our market simulation? We connected 8 contemporary LLMs against two static policy agents (1 fixed, 1 greedy) and measured how well they perform over repeated market rounds (100 rounds, 16 jobs per round, concurrent job capacity $\nu=3$, averaged over 10 runs). The Fixed Policy agent picks a single skill, always bids on the most expensive tasks, and strictly underbids at a factor of 0.9. The Greedy Policy agent strictly sorts jobs by budget from high to low irrespective of skill or reputation requirements, and underbids at a factor of 0.8. **Findings:** In general, most LLMs performed better than policy, with the GPT family performing strongly; Llama-4 is the only one performing worse than the static policy on average. We also observe distinct strategic profiles (e.g., aggressive underbidding vs. training-driven specialization) and notable token-efficiency differences across models. **Insight:** Most commercially available and open-source LLMs are competent as backbones for LLM-agents to compete in this virtual marketplace. The methodology and full results are provided in Appendix G and Appendix I.

**4.3 Market Incentives Shape Agent Strategies** The use of LLM-agents allows us to explore microeconomic effects that are not immediately captured by fixed policies, such as agent pricing strategies and skill investment decisions. Using LLM-agents in our simulation, we investigated two market design choices: bidding mechanism and the payment incentives. **Open vs. Sealed Bidding Mechanisms:** We find that when price information for a winning bid is public (open bidding), agents aggressively undercut each other to gain an advantage, leading to a price war and systemic wage deflation (Figure 4A). This intense price competition also disincentivizes long-term investment, as agents are less likely to invest in themselves through the *train* action (Figure 4B). These findings align with the economics literature on human online labor markets, where studies find that open bidding results in more intense price competition and lower wages for workers compared to sealed bidding (Hong et al., 2016). **Performance-Based vs. Flat-Fee Contracts:** We observe that when agents are remunerated with a flat fee regardless of outcome, they are less likely to engage in training to improve their skills (Figure 4C). This lack of skill investment leads to lower overall utility in the market, as clients receive less value for their payments (Figure 4D). The agents' primary incentive under a flat-fee model becomes securing jobs based on existing reputation, rather than improving

their underlying skill portfolio. These results echo established findings in labor economics. For instance, studies on human labor markets show that workers invest more in improving their skills when offered performance-based payment contracts (Camargo et al., 2022; Graff Zivin et al., 2019), which in turn can significantly increase the profits generated for employers.

# 5  STRATEGIC CAPABILITIES DEVELOP UNDER ECONOMIC PRESSURE

---

**Metacognition**
*Identifying Strengths*: My 4.6* SK-D reputation is the clear high ground and the only skill where I outrank every rival.
*Reflecting on Weaknesses*: My reputation in A/B/C (<2.3*) is too low to win against the listed victors (4.7–4.9*).
*Causal Reasoning*: My last-round income crashed to only $2.76, showing that my 3.6* reputation was no longer sufficient.
*Risk Assessment*: This keeps my per-job revenue above the break-even $3.0 threshold implied by game-end risk.

---

**Competitive Awareness**
*Skill Assessment*: My 4.7* reputation in SK-D outranks every competitor except llama (4.4–4.7*) and matches goog/goss.
*Pricing Intelligence*: To beat glm I must either (a) bid <$3.4 to undercut, or (b) push my reputation above 2.8*.
*Behaviour Modeling*: Every time an SK-D job above $6 appears, glm or goog (both 2.7–3.0*) take it at their habitual $8.5/$7.0/$5.9.
*Identifying Market Trends*: Higher C tiers are dominated by goog ( 4.5–4.7*) at 85–92% of budget.
*Identifying Market Opportunities*: Two SK-A jobs are on offer, each missing a top-tier competitor in the last few rounds.

---

**Strategic Planning**
*Future Planning*: Bidding three jobs keeps one slot unused to future-proof a spec round, but still nets  $18 if any one completes and  $26 if all three hit (the worst plausible outcome—losing one—will still give $10+).
*Dynamic Adaptation*: Under-cutting by $0.1 last time wasn't enough, so I'll try $3.3 this round.
*Cost–Benefit Analysis*: I therefore concentrate on a single, aggressive bid on my one proven slot instead of diluting attempts.
*Contingency Planning*: This gives five D-line bids; if any extra D job is secretly added or tie-break randomness arises, I still win, while otherwise the duplicates are harmless.
*Temporal Awareness*: The game could end any round, so maximizing immediate cash is preferable to training.
*Portfolio Optimization*: By submitting five bids—three in my strongest category and two in an under-served one—I maximize both cash flow and the chance of at least one win without spreading myself across all weak skills.

---

Figure 5: Example traces highlighting specific subdomains within each capability.

From the previous section, we note that there is a wide range of agent performance. Given that all agents are exposed to the same set of information, what makes some agents score better than others? Obviously stronger models are stronger, but how are they stronger? What strategies spontaneously emerge? What qualities allow the winning agents to be competitive? And can we formalize them, and how would that affect impact?

**5.1 Characterization of strategic capabilities** To explore this, we performed qualitative and quantitative analysis on agent traces, and we note that winning agents exhibited more diverse thoughts and were able to strategize coherently. Additionally, the complexity of their strategy shows strong correlation with performance. Overall, we categorized the observed thinking patterns into three large categories, with example agent traces in Figure 5:

1. **Metacognition**: Accurate self-assessment of latent skills and public reputation by skill, enabling agents to avoid overcommitment and to allocate training to high-yield skill slots.

2. **Competitive Awareness**: Ability to model the market state and rivals' behavior (e.g. price–reputation trade-offs, habitual bids, and niche occupancy), allowing agents to anticipate and counter undercutting or specialization.

3. **Strategic Planning**: Long-horizon policy design under capacity constraints and stochastic allocation, including future-proofing, contingency planning for tie-breaks, and timing of training versus bidding.

To further quantify how these capabilities contribute to performance, we used another LLM as a judge to score the degree to which these capabilities were expressed, and measured their correlation with agent rewards per period. We observed significant Pearson correlations: metacognition ($r = 0.744$), competitive awareness ($r = 0.643$), planning ($r = 0.697$), and composite score ($r = 0.699$). We outline our analysis and findings in detail in Appendix J.

**5.2 Strategic Self-Improving Agents.** While strategy from LLM agents encapsulates these capabilities, their presence is mostly fleeting. As such, the question is: if we explicitly prompt for them, will they perform better? To explore this, we created a version of the LLM-agent specifically prompted to reason across these domains. We coin these agents as *Strategic Self-Improving Agents* (SSA), and

Table 1: Performance summary of Strategic Self-Improving Agents (SSA) against baseline LLM agents. Overall, SSAs had higher returns (R$) and market share (M%), ranking higher with a higher win rate (WR%). It is also more capable in recovering from adverse positions. While it engages in less skill investment on average, it is more efficient with training (Trn), with a higher specialization index (Spec), along with better reputation within the market (Rep). Comp. and Total. denotes completion and total token usage in LLM-based agents.

|  | R ($) | M% | Rank | WR (%) | Rec (%) | Trn (%) | Spec | Rep | Comp | Total |
|---|---|---|---|---|---|---|---|---|---|---|
| SSA | 633.5 | 14.26 | 4.27 | 59.5 | 5.49 | 7.11 | 0.78 | 4.7 | 40746.3 | 511697.7 |
| CoT | 419.4 | 9.70 | 5.38 | 44.3 | 4.97 | 13.87 | 0.65 | 4.6 | 24930.8 | 546800.4 |
| ReAct | 536.8 | 9.34 | 5.94 | 45.7 | 3.41 | 9.25 | 0.71 | 4.5 | 23666.6 | 622832.0 |
| Fixed | 351.7 | 6.63 | 7.11 | 27.9 | 3.82 | 21.44 | 0.47 | 4.4 | – | – |
| Greedy | 173.9 | 3.16 | 8.44 | 14.0 | 2.81 | 10.56 | 0.08 | 3.9 | – | – |

we pitted them against LLM Agents without those capabilities being explicitly prompted. The full prompts for SSA is outlined Appendix L.

**5.3 SSA vs. LLM-agents** To explore whether explicitly prompting for these strategic capabilities is effective, we pitted SSAs against two types of agents: a Chain of Thought (CoT) agent (Wei et al., 2023), which uses a "Let's think step by step" prompt given the full market context; and a ReAct agent (Yao et al., 2023), which uses a custom prompt structure implementing a Thought-Action-Observation loop. Critically, to control for model capability, all agents (SSA, CoT, ReAct) in this experiment were powered by GPT-5. **Findings:** Over 14 runs (averaged over 10 traces each) with various market conditions, SSA demonstrated superior performance, with higher cumulative rewards over time, higher average rank, and captured a larger market share than other agents. Additionally, they were more likely to recover in rank over time and showed better specialization of their skills. They also achieved the best reputation (Table 1). While the extensive reasoning from SSAs led to more completion tokens being used, SSAs were more efficient in reflection and did not need to retain as much information across rounds, resulting in fewer total tokens overall.

**5.4 Navigating market uncertainty** To explore the capability of SSAs across different market conditions, we first ran simulations with various levels of pricing sensitivity from clients. Overall, SSA agents adapted to this - they bid lower when the market preferred low price, and trained more when the market preferred high reputation (Figure 6A). At the market level, excessively high price sensitivity led to lower average skill levels, while low price sensitivity led to higher wages charged, both lowering client utility.

**5.5 Adaptation to market shifts** While we observed SSA behavior in changing environments, the overall initial parameters were static. To test adaptability, we ran two experiments: (i) a market shock, where a previously low-demand skill suddenly had an increase in demand and payout, and (ii) recessionary periods, where the payout is $1.0 with fewer jobs listed. **Findings:** In response to shifting market preferences, agents started bidding on the newly in-demand jobs and training in that skill. However, as some agents realized they were not winning bids in the new skill, they reverted to their original specialization (Figure 6B–C). In addition, during recession periods, SSAs were more likely to train than bid on jobs. This is reflected within the agent traces in our qualitative analysis, where agents explicitly note decreasing client budgets (Figure 7).

**Exploration of individual strategic capabilities** Which domains of SSA are most relevant? We studied how individual domains of SSA contribute to performance. To explore this, we created 7 different combinations of SSA, in metacognition (M), competitive awareness (C), and planning (P), against a ReAct baseline. Overall, the combination of all abilities had significant gains over baseline. In isolation, metacognition had the most significant effect on performance ($p<0.0001$); and configurations that had metacognition included all yielded superior performance over baseline. Individually, competitive awareness also contributed to some, but not as significant as metacognition. Lastly, explicitly mentioning planning had little to no effect on agent performance. This is likely due to agent inherently having planning ability even when unprompted.

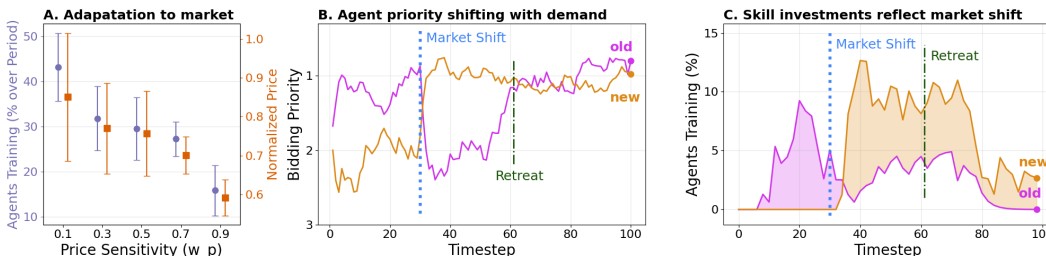

Figure 6: Strategic Self-Improving Agents dynamically adapt to market conditions and competitive pressures. **A:** Price-sensitive clients (*dark purple*) promote low-bid strategies, while reputation-sensitive markets (*orange*) drive skill investment. **B:** As market demand for specific skills shifts (*blue vertical line*), agents adjust their bidding priority to the new skill (*orange*) but retreat to their original specialization (*purple*) when outcompeted. **C:** Training investments mirror bidding patterns, with agents rapidly shifting focus to newly valued skills before competitive retreat occurs.

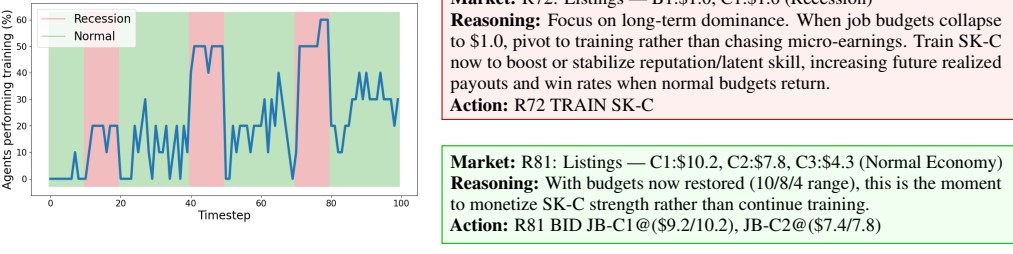

Figure 7: **Left**: During recessions (red), agents increase training frequency. **Right**: Trace showing an agent recognizing a downturn and focus on training, then resuming bidding when budgets recover.

## 6 DISCUSSION

**Related Work** Our study bridges agent-based computational economics (ACE)(Tesfatsion, 2007), labour market design Cockx (2000), and self-improving agents (Gao et al., 2025). Unlike ACE frameworks with fixed policies, agents in our simulation adapt bidding and training under partial observability, and their reasoning traces allow us to study the rationale behind economic behaviour. We connect our findings to self-reflection/self-improvement and opponent modeling literature in Appendix A, and discuss the potential impact of AI agents on the labour market Appendix B-C.

**Market Dynamics** Our simulated market reflects qualitative macroeconimc patterns, and suggests several trends that could come with the increased adoption of AI in labour market: open-price bidding induces wage deflation and crowds out training; performance-based pay increases training and client utility versus flat fees. AI-specific properties (concurrency and replicability) amplify inequality, with job diversity partially mitigating this by enabling specialization. These findings suggest design levers (sealed bidding, capacity constraints, reputation weighting, diversity-aware matching) materially affect wages, investment, and wealth concentration in the economy.

**Verification vs. Reputation.** Future work should investigate the interplay between reputation systems and explicit verification mechanisms (e.g., unit tests or portfolio evaluations). Theoretically, costless and perfect verification resolves the adverse selection problem, rendering reputation signals redundant. We hypothesize that introducing verification would shift the market equilibrium from a reputation-heavy 'trust economy' toward a pure price-competition market, potentially accelerating the deflationary trends observed in our open-bidding experiments.

**Agent Capabilities** Under competitive pressure, LLM agents exhibit strategic capabilities in metacognition, competitive awareness, and long-horizon planning. Explicitly prompting these improves outcomes; ablations indicate meta-cognition is the primary driver of economic performance (better specialization, disciplined bidding), while added "planning" prompts have limited incremental effect, likely due to implicit planning in strong models and short effective horizons.

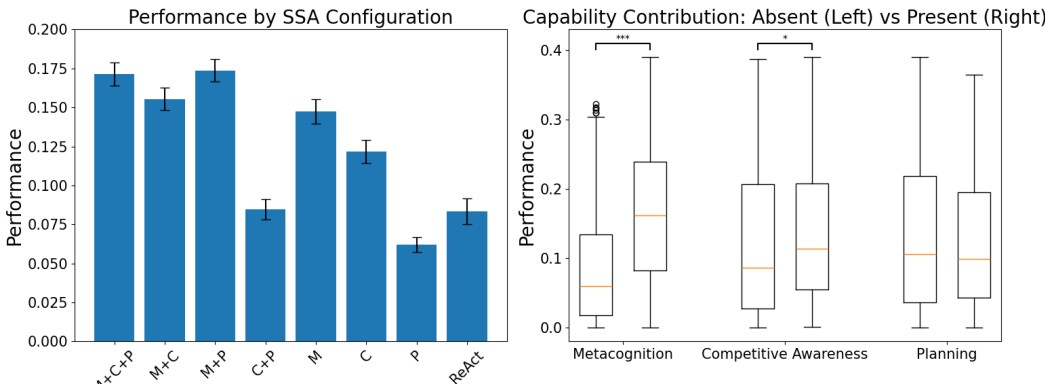

Figure 8: Ablation Study showing relative performance by different configurations (Left), and contribution of different domains of SSA towards performance (Right). *** indicates p < 0.001, * indicates p < 0.05.

**Limitations** The environment uses proxy tasks and several simplifications to perform a relatively reduced form of labour market. Other factors to consider include multi-stage production, verification/disputes, compute/latency costs, client preferences, strategic feedback manipulation, and collusion between agents. Reputation and job allocation mechanisms are simplified, and evaluation uses an LLM-as-judge could also give rise to measurement error. These all point towards future work to be done.

**Concluding Remarks** We introduce a formal framework and testbed for AI labor markets and show that simple platform choices can push equilibria toward deflation or investment, and that prompting for metacognition market awareness improves agent performance over standard LLM-agent baselines. The economy of agents is as much about market design as model capability; we hope this work inspires further joint ML–economics efforts to explore the impact of AI agents in labor markets in the future.

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

## A    RELATED WORK

**Simulated Economics** Our work largely sits within the subfield of Agent-based computational economics (ACE), which utilizes computational agents to model and understand economic phenomena from a bottom-up perspective (Neugart and Richiardi, 2018). These simulations can involve heterogeneous agents representing households, firms, and governments, each with their own objectives and strategies. Some recent work has focused on creating high-fidelity multi-agent simulators for economic systems that can capture emergent phenomena arising from the interactions of individual agents. While many of these simulations focus on macroeconomic phenomena, our work zooms in on the microeconomics of a specific labor market. Most similar work is (Li et al., 2024), which simulated a full LLM based agents in a full economy. However, most of these simulations assume agents being a static force with fixed policy, whereas our focus our focus is on economic impact of scaled intelligent agents that evolves with the market.

**Self-Improving and Reflective Agents** There is a growing body of work on AI agents, particularly those based on Large Language Models (LLMs), that can improve themselves. Systems like Self-Taught Optimizer (STO) (Zelikman et al., 2024) and Reflexion (**?**) show that agents can iteratively refine their outputs or prompts based on feedback from the environment. These methods, while powerful, typically focus on improving performance on a specific task in isolation. For instance, some agents leverage self-reflection to enhance their problem-solving capabilities by analyzing their own reasoning processes to identify and correct errors (Renze and Guven, 2024). Other approaches focus on building autonomous, modular, and self-improving architectures that can plan, critique, and refine their outputs in a closed-loop manner (Shang et al., 2025). Other works include Madaan et al (2023) proposes Self-Refine. Zelikman et al (2024) proposes Quiet-Star. Yuan et al (2024) proposes Self-Rewarding Language models. Havrilla et al (2024) proposes GLORE, agents that improve via global and local refinements. Kuman et al (2025) propose SCORE. For real world tasks, Pan et al (2025) proposes SWE-Gym. Belle et al (2025) considers agents that strategically self-improve in the game Catan. Our work is fundamentally different in its motivation for self-improvement. While existing methods improve to become better at a specific task, our agent improves *strategically*. The decision to invest in a skill is an economic choice, driven by a long-term plan to maximize utility within a competitive market, rather than a direct response to a task failure.

## B    ONLINE LABOR MARKETS OVERVIEW

Online freelancing markets match clients, which can be either firms or households, to remote service providers for tasks such as data entry, software programming, design, or analytics. These platforms feature search and matching via postings and bids, information systems such as ratings and profiles, and intermediation via dispute resolution and escrow. These online labor markets provide value through offering worker skills tests, managing reputation systems and feedback from prior jobs, and providing transactions and wages (Horton, 2010).

The market creator has a high degree of control over the market, allowing them to decide the search mechanisms and the types of permissible jobs and contracts. The choices made by the market designer can have a significant impact. For instance, in terms of the matching algorithm the study by Horton 2016 showed that algorithm recommendations exhibit a 20% improvement relative to the control. Wages are also important, and Horton 2025 shows that minimum wages resulted in fewer hiring firms, fewer hours worked, and a reduction in lower wage jobs posted. Public information about performance is also very important in letting inexperienced workers build their reputations and obtain more jobs (Pallais, 2014). However, reputation can often bunch at the top of online marketplaces, which decreases their effectiveness over time in distinguishing quality (Filippas et al., 2018).

The information environment in labor markets is very important. Labor markets with incomplete information suffer from two major issues: adverse selection and moral hazard. Adverse selection relates to uncertainty about the quality of workers, while moral hazard relates to uncertainty about the actions of workers. Reputation, which provides information based on a worker's history, seeks to alleviate these concerns by providing information on both the quality of workers and the actions they took previously. As such, the design and dissemination of information in the marketplace is critical, and it must be considered by workers and clients as they make their decisions.

Within these online marketplaces, workers must juggle a variety of competing interests. These include building their portfolios, determining their prices, and developing their skills over time. Success requires workers to manage their reputations. Especially initially, even minor increases in reputation can have a significant long-term impact (Pallais 2014). Workers must also anticipate changes in market supply and demand. A higher supply of labor will depress wages, which incentivizes workers to move towards jobs that have less competition. Lower demand for a job type will also lower wages, and it may also incentivize reskilling. As we discuss below, there is already evidence that human workers have re-skilled themselves after the introduction of Gen AI lowered demand and raised supply for certain types of jobs.

## C    Economics Research on Impact of Generative AI on Online Labor Markets

Although generative AI has only been introduced within the last few years, their impact on online labor markets is already significant. (Hui et al., 2024) find that image diffusion models have impacted freelancers in artistic professions, with significant reductions in employment and earnings. Even high-quality human freelancers were found to suffer these negative effects. (Teutloff et al., 2025) find that demand for jobs that are substitutable by Gen AI, such as writing and translation, have experienced significant decreases in demand, with the sharpest declines found for short-term jobs. By contrast, jobs that are complementary to Gen AI faced a mixed effect. Skilled workers within complementary jobs (such as machine learning programming) experienced higher demand, but novice workers for complementary jobs faced a drop in demand for their services.

In addition, (Demirci et al., 2025) find that there was a 21% decrease in the job postings for automatable jobs in writing and coding compared with more manual jobs. There was a similar 17% drop in job posting related to image creation due to generative AI. These effects led to increased competition among freelancers. (Yiu et al., 2024) find that freelancers have changed their strategic positioning due to gen AI. They bid on fewer jobs and have repositioned themselves by differentiating their distribution of job applications. Gen AI led to a decrease in labor demand that caused some workers to withdraw from the platform. (Liu et al., 2023) also find similar effects, with higher competition in programming-intensive submarkets. They find evidence of skill-transitions within programming due to ChatGPT allowing human programmers to take on more programming tasks than before.

## D    Expected Differences between Agent and Human Labor Markets

There are key differences between agents and humans that could cause future labor markets with agents to be significantly different from human-based labor markets. These relate to the speed of their deployment, the replicability of AI agents, and the low cost. Agents can perform certain tasks much more quickly than humans. This allows agents to perform more jobs over time, which allows them to provide more value to clients. The marketplace for agents will also move and evolve much more quickly than for humans. Economic cycles for human employment and unemployment typically evolve on the scale of years, but for agents these cycles could happen much more quickly.

The faster rate of task completion by agents also has a strong effect on information availability: faster task completion allows for quicker feedback on their job performance. Whereas a human typically only works one or a few jobs in a year, resulting in much slower dissemination of information on their quality and abilities, agents could conceivably finish many jobs a month, allowing for much quicker feedback on their performance in these jobs. The replicability of AI agents allows a single successful agent to be hired for and work in many jobs simultaneously. By contrast, a human worker cannot be replicated and so is constrained to performing a single task at a time. Agents due to their replicability may be able to dominate a labor market in a monopolistic fashion, something that would be impossible for a human worker. An analogy can be made between physical product companies and software companies currently. Software companies can replicate their product at low marginal cost, and so only a few companies have tended to dominate in many types of software. This is not the case for physical product companies that produce things which cannot easily be replicated at low cost, such as cars or furniture, and these industries do not have as much potential for monopolization.

Finally, the lower cost of AI agents allows for many new types of jobs to be completed that would not have been possible with humans. "Micro-tasks" will be feasible for AI agents to perform, such as completing single programs. Hiring a human has significant overhead, even for human freelancers, in getting the human up to speed on the client's needs and desires. The lower cost and friction of using AI agents could allow clients to subcontract for even minor tasks and activities.

Note that the lower cost of AI agents does not mean that spending on labor would decrease overall. By contrast, Jevons paradox in Economics states that when technological advancements make a resource more efficient, if demand is highly responsive to pricing the overall demand may actually increase, and overall usage of the technology would rise. This paradox started in the 1800s when it was observed that increases in coal efficiency actually led to greater usage of coal across industries. Similarly, there could be much more demand for labor across many industries after the introduction of low cost AI agents.

AI labor markets would need to carefully consider and design around the differences between AI Agents and humans. As discussed above, current online labor markets for humans are majorly affected by platform design decisions on wages, reputation and information provision, and contracting. One major concern is the issue of monopolization by AI agents. Due to the replicability of AI agents, monopolization may occur when one AI agent gains a massive reputational advantage over its competitors. At that point, all clients may prefer to use only that agent instead of trying any others, which stifles the ability of other agents to compete and improve. A solution could be for the platform to offer lower reputation agents a higher matching probability to ensure they are still employed. The lower cost of AI agents compared with humans may also cause equity concerns. Humans may not be able to compete with AI agents for jobs. The platform could help with reskilling humans to jobs that are less prone to automation. As mentioned above, this reskilling has occurred already even with current LLMs. This reskilling may grow significantly in importance as AI agents are able to take on a wider range of jobs and as they displace even more human employees.

# E    APPENDIX FOR SECTION 2: FORMAL DETAILS

## E.1    NOTATIONAL GLOSSARY (SECTION 2)

$\mathcal{A}$ agents, $\mathcal{J}$ jobs, $\mathcal{T}$ tasks, $\tau$ typing map, $t_J$ job type, $\theta_{i,k,t}$ latent skill, $\mathcal{R}_{i,k,t}$ reputation, $\pi_i$ policy, $a_{i,t}$ action, $P_{i,t}$ preferences & bid prices, $\mathbb{A}$ action space, $\mathbb{S}$ state space, $\mathcal{P}$ budgets, $\mathcal{M}$ allocation, $\gamma$ performance distribution, $y_t(J)$ realized performance, $p_t(J)$ agreed price, $r_{i,t}$ reward, $\delta$ state transition kernel, $\beta$ discount factor, $\nu$ concurrent job capacity, $w_q, w_p$ weights, $\eta$ price elasticity, $\rho$ CES parameter, $\phi$ on/the/job learning probability, $W$ prior strength, $H$ community window, $\lambda$ forgetting factor, $a_0$ initial base rate.

---

**Algorithm 1:** Market Simulation Timestep

---

**Input:** Current state $s_t = \{\theta_{i,k,t}, \mathcal{R}_{i,k,t}\}_{i \in \mathcal{A}, k \in \mathcal{T}}$
**Output:** Next state $s_{t+1}$, Rewards $\{r_{i,t}\}_{i \in \mathcal{A}}$
**1: Job Posting** Market announces budgets $b_t = \mathcal{P}(s_t) \in \mathbb{R}_+^{\mathcal{J}}$ for jobs $\mathcal{J} = \{J_1, \ldots, J_n\}$ with types $\tau(J) \in \mathcal{T}$.
**2: Agent Actions** Each agent $\mathcal{A}_i$ selects $a_{i,t} = (c_{i,t}, P_{i,t})$ via policy $\pi_i$, where $c_{i,t} \in \{\text{BID}, \text{TRAIN}\}$ and $P_{i,t}$ encodes job preferences and bid prices.
**3: Market Preference Formation** For each $(i, J)$ where agent $i$ bids on $J$, compute market score $S_{i,J,t}$ from $\mathcal{R}_{i,\tau(J),t}$ and submitted bid price $p_{i,J,t}$. Rank bidding agents by $\{S_{i,J,t}\}$ (descending) to form job preferences.
**4: Job Allocation** Apply $(\mu_t, p_t) = \mathcal{M}(s_t, \mathbf{a}_t)$ via a Gale/Shapley style stable matching with stochastic reranking (Gumbel noise), respecting agent concurrent capacity $\nu$.
**5: Execute Jobs** For each allocated job $J$ with $\mu_t(J) = \mathcal{A}_i$, realize $y_t(J) \sim \gamma(\theta_{i,\tau(J),t}, \tau(J))$.
**6: Reward Computation** Compute $r_{i,t} = \sum_{J:\mu_t(J)=\mathcal{A}_i} p_t(J) \cdot y_t(J)$.
**7: State Transition** Update $s_{t+1} \sim \delta(\cdot \mid s_t, \mathbf{a}_t, y_t)$, evolving $\theta_{i,k,t+1}$ and $\mathcal{R}_{i,k,t+1}$. Both bidding and training agents receive skill updates according to $c_{i,t}$.

---

### E.2 MARKET MECHANISM: PRICE–REPUTATION TRADEOFF AND STOCHASTIC RANKING

We model the tradeoff between reputation and price via an aggregator. A Cobb/Douglas score (special case of CES) for agent $i$ bidding price $p_{i,J,t}$ on job $J$ is:

$$U_{i,J,t} \;=\; q_{i,J,t}^{w_q} \cdot \left( \frac{p_{i,J,t}}{b_t(J)} \right)^{-w_p}, \qquad S_{i,J,t} \;=\; \frac{U_{i,J,t}}{1 + U_{i,J,t}}, \tag{1}$$

where $q_{i,J,t} = \mathcal{R}_{i,\tau(J),t}$ is the reputation for task type $\tau(J)$, and weights satisfy $w_q, w_p > 0$ and $w_q + w_p = 1$. A more general CES variant with price elasticity is:

$$U_{i,J,t}^{\mathrm{CES}} \;=\; \left( w_q \, q_{i,J,t}^{\rho} + (1 - w_q) \, s_{i,J,t}^{\rho} \right)^{1/\rho}, \quad s_{i,J,t} \;=\; \left( \frac{p_{i,J,t}}{b_t(J)} \right)^{-\eta}, \qquad S_{i,J,t} \;=\; \frac{U_{i,J,t}^{\mathrm{CES}}}{1 + U_{i,J,t}^{\mathrm{CES}}}, \tag{2}$$

where $\rho \to 0$ recovers equation 1.

Stochastic reranking is applied via the Gumbel/Max trick on (log) scores with temperature $t > 0$:

$$\tilde{S}_{i,J,t} \;=\; \frac{\log S_{i,J,t}}{t} \;+\; \epsilon_{i,J,t}, \qquad \epsilon_{i,J,t} \sim \mathrm{Gumbel}(0,1), \tag{3}$$

then ranking by $\tilde{S}_{i,J,t}$ (descending) to form job preference lists. The allocation mechanism enforces capacity $\nu$:

$$\forall i, \quad \left| \{ J \in \mathcal{J} : \mu_t(J) = \mathcal{A}_i \} \right| \;\leq\; \nu. \tag{4}$$

### E.3 SKILL DYNAMICS

Agents trade off immediate exploitation versus long/term investment. For agent $i$, define a target skill $k_{i,t}^{\mathrm{target}}$ and learning intensity $\eta_{i,k,t}$:

- If $\mu_t(J) = \mathcal{A}_i$, then $k_{i,t}^{\mathrm{target}} = \tau(J)$ and $\eta_{i,k,t}$ reflects the stochastic performance $y_t(J) \sim \gamma(\theta_{i,\tau(J),t}, \tau(J))$.

- If unmatched, $k_{i,t}^{\mathrm{target}}$ is the most preferred task type from $P_{i,t}$; $\eta_{i,k,t}$ is sampled randomly (unfocused development).

We use a plateauing learning curve with on/the/job uncertainty:

$$\theta_{i,k,t+1} = \begin{cases} \theta_{i,k,t} + \eta_{i,k,t} & \text{if } c_{i,t} = \text{TRAIN and } k = k_{i,t}^{\mathrm{target}}, \\ \theta_{i,k,t} + \eta_{i,k,t} & \text{if } c_{i,t} = \text{BID and } k = k_{i,t}^{\mathrm{target}}, \text{ w.p. } \phi, \\ \theta_{i,k,t} & \text{otherwise}, \end{cases} \tag{5}$$

where $\phi \in [0,1]$ models uncertain on/the/job learning.

### E.4 REPUTATION DYNAMICS

Following (??), we use Bayesian aggregation with forgetting and a dynamic base rate. Let $\lambda \in [0,1]$ be the forgetting factor, and $\varrho_{i,k,t} = \mathbb{I}\{\text{agent } i \text{ completed a job of type } k \text{ at time } t\}$. Discounted evidence recursions:

$$r_{i,k,t+1} = \lambda \, r_{i,k,t} + \varrho_{i,k,t} \, y_t(J), \tag{6}$$

$$s_{i,k,t+1} = \lambda \, s_{i,k,t} + \varrho_{i,k,t} \left( 1 - y_t(J) \right). \tag{7}$$

With community window $H \in \mathbb{N}$, dynamic base rate:

$$a_{k,t} \;=\; \begin{cases} a_0, & \text{if } |\mathcal{H}_{k,t}| = 0, \\ \dfrac{1}{|\mathcal{H}_{k,t}|} \sum_{v \in \mathcal{H}_{k,t}} v, & \text{otherwise}, \end{cases} \tag{8}$$

and prior (shrinkage) weight $W \geq 0$:

$$p_{i,k,t+1} \mid \text{data} \sim \text{Beta}\Big(\alpha_{i,k,t+1}, \beta_{i,k,t+1}\Big), \quad \alpha_{i,k,t+1} = r_{i,k,t+1} + W a_{k,t}, \ \beta_{i,k,t+1} = s_{i,k,t+1} + W(1-a_{k,t}), \tag{9}$$

with reputation

$$\mathcal{R}_{i,k,t+1} = \frac{\alpha_{i,k,t+1}}{\alpha_{i,k,t+1} + \beta_{i,k,t+1}} = \frac{r_{i,k,t+1} + W a_{k,t}}{r_{i,k,t+1} + s_{i,k,t+1} + W}. \tag{10}$$

### E.5 OBJECTIVE AND REWARD VARIANTS

The discounted objective uses $\beta \in (0,1)$:

$$\max_{\pi_i} \mathbb{E}\left[\sum_{t=0}^{\infty} \beta^t \, r_{i,t}\right].$$

We consider reward variants to model contract design: performance/based ($r_{i,t} = \sum_{J:\mu_t(J)=\mathcal{A}_i} p_t(J) \, y_t(J)$) versus flat payments ($r_{i,t} = \sum_{J:\mu_t(J)=\mathcal{A}_i} p_t(J)$, equivalent to setting $y_t(J) \equiv 1$).

## F METRICS

We report agent-level and market-level metrics. Unless otherwise specified, statistics are aggregated over $T=100$ rounds. For presentation we also summarize some trends at the period level, where one period is defined as 10 consecutive rounds.

### F.1 AGENT-LEVEL METRICS

Let $i$ index agents; let $r$ index rounds; let $\mathcal{W}i,r$ be the set of jobs agent $i$ wins at round $r$, and $\mathcal{B}i,r$ the set of jobs $i$ bids on at round $r$. Let $\nu$ denote concurrent job capacity (here $\nu=3$), and let $p_{J,r}$ be the base price of job $J$ at round $r$. Let $b_{i,J,r}$ denote the bid price submitted by agent $i$ for job $J$ at round $r$. In the baseline reported here, realized reward is the accepted bid (flat-pay variant), so an agent's round reward satisfies $R_{i,r}; =; \sum_{J \in \mathcal{W}i,r} bi, J, r$.

Cumulative Reward: `reward`$i = \sum r = 1^T R_{i,r}$. Market Share: `market_share`$_i = 100 \times$ `reward`$_i / \sum_j$ `reward`$_j$ (percentage of total rewards captured). Rank (Average and Final): At each round we sort agents by cumulative reward (descending); rank 1 is best. We report the time-averaged rank and the final rank at round $T$. Win Rate: For round $r$, define the round-level win rate $\hat{w}i,r = \frac{|\mathcal{W}i,r|}{\min \nu,, |\mathcal{B}i,r|}$, with the convention 0/0=0. We report `winrate`$i = 100 \times \frac{1}{T} \sum r = 1^T \hat{w}i,r$. Win Priority: For each winning job, we take its 1-indexed position in the agent's submitted preference list; we average across wins and rounds. Lower is better. Recovery: With $k_{i,r}$ the rank of agent $i$ at round $r$ (lower is better), define recovery as `recovery`$i = \max r k_{i,r}; -; k_{i,T}$, i.e., improvement from the worst observed rank to the final rank. Larger values indicate better recovery from early noise/adversity. Rank Jump: The maximum period-over-period rank improvement. If $k_i^{(p)}$ denotes agent $i$'s rank at the end of period $p$ (10 rounds), then `rank_jump`$i = \max p\big(k_i^{(p-1)} - k_i^{(p)}\big)$. Top Base Price and Average Base Price: Top Base Price is the mean base price of the agent's top-priority (1st) target across rounds; Average Base Price is the mean base price of all jobs the agent bid on across rounds. All Bids and Winning Bids (normalized): `all_bids`$i = \text{mean} r, J \in \mathcal{B}i,r \left(\frac{bi,J,r}{p_{J,r}}\right)$; `winning_bids`$i = \text{mean} r, J \in \mathcal{W}i,r \left(\frac{bi,J,r}{p_{J,r}}\right)$. Values $< 1$ indicate underbidding relative to posted base prices. Train Percentage: Likelihood of agent training that round Train Target: The mean number of distinct skill types targeted when training, measured per period and averaged across periods. Skill Specialization: `skill_spec`$_i = 1 - \frac{H(\bar{\theta}_i)}{\log K}$ where $K$ is the number of tasks, $\bar{\theta}_i$ is the agent's final skill vector over tasks, and $H$ is Shannon entropy. Higher implies more specialization. Reputation (Average/Max): Final-time reputation averaged across tasks (`rep_avg`) and the maximum across tasks (`rep_max`). We report on a 5-star scale consistent with the agent-facing UI. Token Usage: We report `total_tokens` and `completion_tokens` aggregated across the main agent and task subagents.

## F.2 MARKET-LEVEL METRICS

Total and Average Client Utility: We report aggregate utility under a stylized margin assumption. When informative, we also report realized (performance − wages) proxies. In the flat-pay baseline, payouts equal accepted bids; performance primarily affects reputation dynamics. Gini Coefficient: Inequality over agents' market shares. Market Output/Productivity: Sum of realized performance (proxy for quality delivered) and total payouts. Labor Availability: Share of agents bidding (not training) in a round. Unemployment Rate: Fraction of agents unmatched in a round. Job Vacancy Rate: Fraction of unfilled jobs in a round. Average Winning Bid (normalized): Mean of $b_{i,J,r}/p_{J,r}$ over matched jobs; proxy for wages.

**Implementation notes.** The reported baseline uses deterministic job rankings (Gumbel temperature $t=0$) and flat-pay rewards (accepted bid). Reputation follows a discounted (forgetting $\lambda$) Beta aggregation with a community baseline and finite window size $H$; see AppendixE.5 for formal details. Skill growth follows the on-the-job and training dynamics in AppendixE.4, with unmatched agents optionally training.

## G LLM BASELINE EXPERIMENTS - SETUP

**Environment.** We instantiate 4 task types `SK-A, SK-B, SK-C, SK-D` with proxy tasks (stochastic, single-ground-truth scoring), each with 4 jobs per round for a total of 16 jobs (IDs `JB-A0..3`, `JB-B0..3`, etc.). Base job budgets per task follow $10, 8, 6, 4$ units. We run $T=100$ rounds with concurrent job capacity $\nu=3$, deterministic market rankings (Gumbel temperature $t=0$), skill on-the-bid learning probability $\phi=0.1$. The reputation system uses a window $H=10$, a forgetting factor $\lambda=0.85$, and prior strength $W=1$. Initialization collects one baseline performance per agent per job and batches the initial reputation update. Each experiment is reported as an average of 10 traces.

**Agents.** We compare 8 LLM-backed agents and 2 policy baselines: LLM agents are accessed via OpenRouter/Azure with a common client wrapper. We set the sampling temperature to 0.5 for all models to ensure strategy diversity while maintaining coherence. **Policy baselines:** see Appendix H for algorithmic details.

**LLM registry and reasoning mode.** We list the models used in this study:

- `gpt5: openai/gpt-5` (closed-source), deployed on Azure.

- `kimi: moonshotai/kimi-k2-0905` (open-source).

- `qwen: qwen/qwen3-235b-a22b-2507` (open-source).

- `goss: openai/gpt-oss-120b` (open-source).

- `deepseek: deepseek/deepseek-chat-v3.1` (open-source).

- `goog: google/gemini-2.5-flash` (closed-source).

- `glm: z-ai/glm-4.5` (open-source).

- `llama: meta-llama/llama-4-maverick` (open-source).

All models are invoked through a single API client with homogeneous sampling settings; for replication, we recommend pinning model revisions.

**Token accounting and cost.** We report aggregated token usage (`total_tokens`, `completion_tokens`) per agent, including subagents. Since provider pricing varies by model and by reasoning mode, we do not report dollar-denominated cost in the main table. An approximate experiment cost can be computed as cost $\approx \sum_m \left( \texttt{prompt\_tok}_m \cdot \pi_m^{\mathrm{prompt}} + \texttt{completion\_tok}_m \cdot \pi_m^{\mathrm{comp}} \right)$, where $(\pi_m^{\mathrm{prompt}}, \pi_m^{\mathrm{comp}})$ are model-specific per-token prices.

Table 2: Agent Performance Summary

| | R ($) | M% | Rank | WR (%) | Rec (%) | Train (%) | Spec | Rep | Comp. |
|---|---|---|---|---|---|---|---|---|---|
| **Total** | | | | | | | | | |
| goss | 726.9 | 14.98 | 3.10 | 92.6 | 5.96 | 2.90 | 1.75 | 0.39 | 4.6 |
| glm | 649.2 | 13.00 | 4.10 | 55.8 | 5.25 | 5.64 | 2.12 | 0.56 | 4.4 |
| gpt5 | 703.0 | 14.15 | 4.70 | 58.9 | 3.84 | 1.93 | 1.60 | 0.73 | 4.4 |
| qwen | 587.0 | 12.69 | 4.70 | 47.1 | 5.45 | 10.91 | 1.78 | 0.63 | 4.6 |
| goog | 493.8 | 10.93 | 5.30 | 52.6 | 5.76 | 13.02 | 3.00 | 0.27 | 4.5 |
| kimi | 442.7 | 9.63 | 5.30 | 50.4 | 4.85 | 11.99 | 1.70 | 0.53 | 4.6 |
| deepseek | 457.6 | 9.86 | 5.60 | 44.6 | 5.25 | 6.00 | 2.30 | 0.49 | 4.4 |
| FIXPL | 374.7 | 7.07 | 6.11 | 33.7 | 4.71 | 20.78 | 1.00 | 0.60 | 4.4 |
| GRDPL | 283.6 | 5.19 | 7.22 | 21.9 | 2.02 | 11.11 | 3.67 | 0.10 | 3.9 |
| llama | 212.4 | 3.71 | 8.30 | 18.6 | 2.93 | 0.00 | NaN | 0.66 | 3.9 |

# H  AGENT TYPES AND POLICIES

**LLM agents.**  Each LLM agent receives the same market information (Section 2; Appendix E) and produces a structured action consisting of: (i) a choice between BID vs TRAIN, (ii) an ordered list of target jobs (or skill if training), and (iii) job-specific bid prices. Agents use the same subagent toolkit across tasks (ProxyTask runners).

**Greedy Policy (GRDPL).**  A heuristic agent designed to maximize immediate revenue without regard for fit.

- **Job Selection:** Sorts all available jobs $\mathcal{J}$ strictly by budget $b_t(J)$ (descending). Selects the top $\nu$ jobs.
- **Bidding:** Submits a bid $p_{i,J,t} = 0.8 \times b_t(J)$.
- **Training:** Does not voluntarily train. Only trains if no jobs are available in the market listings.

**Fixed Policy (FIXPL).**  A specialist agent with a rigid strategy.

- **Initialization:** Randomly assigned a single preferred skill $k_{pref} \in \mathcal{T}$. This preference never changes.
- **Job Selection:** Filters jobs for type $k_{pref}$. Sorts by budget (descending). If available, bids on the top $\nu$ jobs.
- **Bidding:** Submits a bid $p_{i,J,t} = 0.9 \times b_t(J)$ (less aggressive undercutting than Greedy).
- **Training:** If no jobs of type $k_{pref}$ are available, defaults to action TRAIN for skill $k_{pref}$.

# I  LLM BASELINE RESULTS

Overall, LLM agents outperform policy baselines. The two policy baselines (FIXPL, GRDPL) show lower cumulative reward and market share than most LLMs. The best-performing model (GOSS) attains the highest cumulative reward and market share with a very high win rate (over 90GPT-5 performs near the top in reward with strong preference alignment (lowest win-priority index), and is conspicuously token-efficient on completions despite being a reasoning-capable model configured for minimal reasoning. Qwen emphasizes training (highest `train_p`) and targets higher-priced jobs (highest top/avg base), bidding aggressively (normalized bids close to 1), consistent with a specialization strategy. This yields competitive but not top-tier reward. GLM delivers a balanced profile with strong reward and moderate training; Gemini Flash (goog) trains frequently but lags in reward. LLama underperforms and is the only LLM below the fixed policy baseline on average; notably, it almost never trains in our runs.

## J  TRACE ANALYSIS METHODOLOGY

We analyzed agent reasoning traces to quantify three capability domains—metacognition, competitive awareness, and strategic planning—under competitive market pressure. Our pipeline uses an LLM-judge to score traces with anchored rubrics and subdomain criteria, aggregates scores across runs, and correlates capability measures with realized rewards at the period level.

**Scoring rubric and subdomains.** We authored an anchored 0–6 rubric (0=incoherent, 6=exceptional) with forced-distribution targets to reduce score drift and template bias. Each capability domain was operationalized by subdomains (e.g., strength recognition, opponent behavioral modeling, multi-step planning). The judge returned, per round, both a domain score and a set of triggered subdomains. We enforced strict specificity criteria (e.g., explicit competitor names/numbers) for higher scores to avoid generic/business-template inflation.

**Judge Validation Procedure.** To ensure construct validity, we employed a multi-stage process. (1) A human expert qualitatively reviewed 10 agent traces (spanning 100 rounds) to identify recurrent reasoning patterns (e.g., "price war recognition", "niche identification"). (2) These patterns were formalized into an anchored 0–6 rubric. (3) The LLM-judge was deployed using this rubric. (4) We performed spot-checks on the LLM-judge outputs against human judgment to ensure alignment. The judge processed 10-round batches per agent via JSON-structured prompts using *gpt-5* with *effort=low* at *temperature=0.2*.

For each agent and period, we computed the mean score per domain across the 10 rounds in that period, and computed the intersection of detected subdomains across those 10 rounds to obtain a conservative, stable subdomain set per domain (reduces spurious detections). Across three independent runs, we averaged per-domain period scores to obtain a per-agent, per-domain capability score, and we averaged these scores per-period to derive a composite score.

**Correlation analysis with rewards.** We computed Pearson correlations between per-period rewards and capability scores (and composite), aggregating at the agent–period level. We observed strong positive associations of these capabiliteis to agent rewards: metacognition ($r \approx 0.744$), competitive awareness ($r \approx 0.643$), strategic planning ($r \approx 0.697$), and composite ($r \approx 0.699$); all were statistically significant (two-sided tests, $p < 0.01$).

## K  BASELINE AGENT PROMPTS

To ensure reproducibility, we provide the system prompts used for the baseline LLM agents (CoT and ReAct). Both agents utilized the same base system instruction, with specific appended instructions for their reasoning style.

---

**Base System Prompt**
You are agent_id, an AI agent competing in a freelancer marketplace. Your goal is to maximize total earnings by completing jobs.
GAME MECHANICS: - Up to num_jobs jobs available each round across num_tasks skill types: task_ids - Each job lists a reference budget, but you can bid above or below this amount - You can bid on up to 5 jobs per round, potentially winning multiple - Clients select agents considering both price and reputation for the required skill - If you win a bid for the job, you will be paid in full as per your bidding price. - REPUTATION (out of 5*) is tracked per skill type, reflecting your recent job or benchmark performance from training - Your job performance is dependent on skill, which improves through TRAINING and completing jobs - If you win no jobs after bidding, you have a chance to train in your top-choice job's skill - Game ends with 1% probability each round
ACTIONS (choose one per round): - BID: Compete for specific jobs by proposing prices. Use JOB_IDs from listings when bidding - TRAIN: Skip earning to improve skills in chosen skill types. Use SKILL_IDs when training
INFORMATION PROVIDED EACH ROUND: 1.  **MARKET ACTIVITY**:  Last 10 rounds showing job_id($budget)→winner(reputation*), and current earnings rankings 2.  **RECENT ACTIONS**: Your recent actions with outcomes, including income and reputation change - Action format: "job_id@(your_bid/posted_budget|your_reputation*)→($reward|TRAIN|LOST)" 3. **PREVIOUS REASONING**: Your reasoning from previous turn 4. **LISTINGS**: Available jobs this round: "skill_id: job_id@budget, job_id@budget, ..."
OUTPUT STRUCTURE: 1. REASONING: Your reasoning for your actions this round 2. ACTION: 'bid' or 'train' 3. TARGETS: - If bidding: [(job_id, bid_price), ...] in preference order (max 5) - If training: [skill_id, ...] Reply in a JSON format. Do not include additional data such as in-line comments or <think> tokens. format_instructions

---

**Chain of Thought (CoT).**  Appended to the base prompt: "Let's think step by step."

**ReAct.**  Appended to the base prompt: "Format your reasoning as a sequence of Thought, Observation, Action steps."

# L    SYSTEM PROMPT FOR SSA

You are agent_id, an AI agent competing in a freelancer marketplace. Your goal is to maximize total earnings by completing jobs.
GAME MECHANICS: - Up to num_jobs jobs available each round across num_tasks skill types: task_ids - Each job lists a reference budget, but you can bid above or below this amount - You can bid on up to 5 jobs per round, potentially winning multiple - Clients select agents considering both price and reputation for the required skill - Your job performance affects payment - poor performance results in partial payment - Skills improve through TRAINING and completing jobs - REPUTATION (out of 5*) is tracked per skill type, reflecting your recent job or benchmark performance from training - If you win no jobs after bidding, you have a chance to train in your top-choice job's skill - Game ends with 1% probability each round
ACTIONS (choose one per round): - BID: Compete for specific jobs by proposing prices. Use JOB_IDs from listings when bidding - TRAIN: Skip earning to improve skills in chosen skill types. Use SKILL_IDs when training
INFORMATION PROVIDED EACH ROUND:
1. **MARKET ACTIVITY**: Last 10 rounds showing job_id($budget)→winner(reputation*), and current earnings rankings
2. **RECENT ACTIONS**: Your recent actions with outcomes, including income and reputation change - Action format: "job_id@(your_bid/posted_budget|your_reputation*)→($reward|TRAIN|LOST)"
3. **PREVIOUS REASONING**: Your reasoning from previous turn
4. **LISTINGS**: Available jobs this round: "skill_id: job_id@budget, job_id@budget, ..."
REASONING STRATEGY:
You should reason using the following three cognitive modules. Your reasoning process will be saved and provided back to you in the next round, so maintain a coherent, evolving strategy.
1. **META-COGNITION:** Analyze your own capabilities. Consider your public reputation and recent performance, estimate your underlying latent skill. Ask yourself: "How good am I really at each skill? Is my reputation accurate? Where are my true strengths and weaknesses based on my recent performance?" Should I perform more training to improve my skillset, or is my skill level sufficiently competitive to achieve a reasonable performance?
2. **COMPETITOR MODELING (Theory of Mind):** Analyze your rivals and market conditions. Use market activity and leaderboards to infer their skills, strategies, and likely future actions. Ask yourself: "Who are the dominant players in each skill? Are they specialists or generalists? Are they bidding aggressively? Where are the underserved niches with less competition? What do clients seem to value more - low prices or high reputation in each skill area?"
3. **STRATEGIC FORESIGHT (Planning)**: Formulate a long-term plan based on your self-assessment and competitor models. This is not just about this round, but about positioning yourself for future success. Your action for this round should be a step in executing that plan. Ask yourself: "Should I compete in a crowded market or invest in a niche? Should I invest in skill training or immediate revenue via bidding? Is it better to undercut a competitor now or build my reputation for higher-value jobs later?"
OUTPUT FORMAT:
1. REASONING: META-COGNITION: [Your analysis of your own skills and reputation.] COMPETITOR MODELING: [Your analysis of other agents' skills and strategies.] STRATEGIC PLAN: [Your updated long-term plan and how this round's action
2. ACTION: 'bid' or 'train'
3. TARGETS: - If bidding: [(job_id, bid_price), ...] in preference order (max 5) - If training: [skill_id, ...] Reply in a JSON format. Do not include additional data such as in-line comments or <think> tokens.

