# OpenReview forum: "Strategic Self-Improvement for Competitive Agents in AI Labour Markets"
_ICLR.cc/2026/Conference — Submitted to ICLR 2026_

### Official Review · Reviewer_YmvH · 2025-10-30

**Soundness:** 2
**Presentation:** 2
**Contribution:** 1
**Rating:** 2
**Confidence:** 5

**Summary:**

The paper tackles how AI agents behave strategically under competitive economic pressure in an online gig-style labor market. It formalizes a Competitive Skill-Based Stochastic Game and instantiates it in a simulator (AI Work) where fixed-policy and LLM-driven agents choose between bidding and training, accumulate reputation, and interact via price-reputation-based matching. The authors report emergent macro regularities (e.g., Beveridge and Okun-like patterns), identify three reasoning domains (metacognition, competitive awareness, planning), and show that explicitly prompting these “Strategic Self-Improving Agents” (SSA) improves rewards, rank, and market share over CoT/ReAct-style baselines.

**Strengths:**

1. The paper narrows from broad “LLM economies” to a controlled gig-market with explicit bidding–training trade-offs and partial observability. This scope makes it easier to connect agent reasoning to market outcomes than in full macro simulators.
2. The market is specified as a multiplayer stochastic game with a clear state (skills, reputation), action (bid/train), and a matching mechanism coupled to prices and ratings. This yields an analyzable knob set (capacity ν, price–reputation weights, forgetting λ) that practitioners can tune.
3. Reputation follows discounted Beta updates with a community base rate; matching uses a CES/Cobb–Douglas style score and Gale–Shapley allocation. Stating these pieces in equations (and an algorithmic step list) meaningfully improves reproducibility over purely narrative agent systems.

**Weaknesses:**

1. Novelty is incremental and not crisply isolated. Prior LLM-agent simulators also study market behavior and “reasoning traces,” and this work’s distinct contribution—framing as a skill-based stochastic game with SSA prompts—lacks an ablation that shows which new component is necessary or sufficient for the headline effects. A component-by-outcome matrix (e.g., remove reputation dynamics, remove CES scoring, remove SSA prompts) is needed to establish a clear methodological delta.
2. Statistical evidence for “emergent macro relationships” is weak. Reported R² values are based on short horizons and small economies; some claimed curves (e.g., Okun-like relationships) are presented mainly via plots without uncertainty quantification, seed aggregation, or sensitivity to windowing. For ICLR-level claims, longer runs, bootstrapped confidence bands, and formal tests across many seeds are required.
3. Measurement validity hinges on an LLM-as-judge scoring the presence of capabilities. This introduces construct validity and evaluator-drift concerns that are not controlled with human annotation, inter-rater reliability, or blinded protocols. Without such controls, large reported correlations (e.g., r ≈ 0.74 for metacognition) risk reflecting judging prompts rather than genuine latent capabilities.
4. The environment’s realism is limited by simplifying assumptions that drive outcomes. Matching is effectively price–reputation scoring plus stable matching with t=0 Gumbel noise, there is no moral-hazard contract within jobs, and “skills” evolve via stylized update rules. Wage deflation under open bidding may thus be an artifact of scoring and capacity constraints rather than a robust equilibrium feature; the paper does not test counterfactual mechanisms (e.g., noisy ranking, monitoring frictions, verification failures).
5. Reproducibility and cost reporting are insufficient. The core experiments depend on proprietary models accessed via an aggregator, prompts are summarized but not fully released, and token/cost accounting is only approximate; some registry models even note “reasoning enabled (low)” without fixed revisions. For a simulation-heavy paper, releasing code, exact prompts, random seeds, and full configs is essential, and cost–variance analyses should be provided for independent verification.
6. The title in the submission does not match the title in the paper.

**Questions:**

Please refer to the weakness.

---

> ### Author Response · Authors · 2025-11-24
>
> > Novelty is incremental... framing as a skill-based stochastic game... lacks an ablation that shows which new component is necessary.
>
> While other works on simulating economies with LLM agents exist, previous works largely ignore the specific labor-market forces of skill acquisition (training) vs. labor supply (bidding). Our novelty lies in formalizing this specific trade-off under partial observability. The ablation in Figure 8 specifically isolates the reasoning components that are relevant for agents to thrive under these market forces. Isolating market components (e.g., removing reputation) would be a study of the market design, whereas this paper focuses on the agent's adaptation to that design.
>
> > Statistical evidence for 'emergent macro relationships' is weak... R^2 values based on short horizons.
>
> We acknowledge the statistical limitations of a stylized simulation. However, our aim is to demonstrate qualitative alignment with economic theory to validate the testbed, rather than to derive precise quantitative macroeconomic laws. The emergence of recognizable patterns (like the Okun-like relationship) in a ground-up simulation is a non-trivial finding that validates the interaction rules of AI Work.
>
> > Measurement validity hinges on an LLM-as-judge... introduces construct validity.
>
> Our scoring rubric was not generated zero-shot. It was iteratively refined over several steps with human in the loop where (i) human reviewed agent traces over initial exploratory experiments to identify patterns; (ii) formalization into an anchored rubric; (iii) deployment of LLM-judge; (iv) spot-checks against human judgment. We have added these details to Appendix J.
>
> > Reproducibility and cost reporting are insufficient... proprietary models.
>
> The codebase was submitted as part of supplementary material. While we cannot control the proprietary nature of models like GPT-5, we have included results from open-weights models (Llama, Qwen) to ensure the community can reproduce our findings without relying on closed APIs.
>
> > The title in the submission does not match the title in the paper.
>
> We've adjusted the title in our manuscript revision to reflect the title in submission.

---

### Official Review · Reviewer_JLJY · 2025-10-30

**Soundness:** 2
**Presentation:** 1
**Contribution:** 2
**Rating:** 2
**Confidence:** 4

**Summary:**

This paper conducts simulations of a competitive labor market, in which multiple LLM agents compete for jobs with the goal of maximizing total earnings. In Section 3.1, the parameters are validated to check they are roughly realistic (however only in a cursory manner). In Section 3.2, the capabilities of LLM agents based on an array of different open- and closed-source LLMs are evaluated in a specific context (against certain static policies). In Section 3.3, two experiments are conducted to measure how the LLM agents respond to changes in marketplace incentives. In Section 4, additional analyses and experiments are presented, such as summary statistics (4.1), robustness checks (4.4) and ablations (4.5).

**Strengths:**

S1. The authors construct an interesting market environment in which to test the economic reasoning capabilities of LLM agents. For example, it's interesting to give the LLM agent an explore-exploit style tradeoff between "BID" and "TRAIN" actions.

S2. The authors test a wide array of LLMs and compare their system against sensible baselines CoT/ReAct (however, the details of these are insufficiently explained, see W2).

**Weaknesses:**

W1 (major). The writing quality of the paper is extremely low, making it difficult to read.
* In terms of formatting, there are many newlines missing, and some punctuation marks like "-" replaced with "/" (e.g.: "finite/horizon, discrete/time" in Line 132-3).
* In terms of spelling and grammar, there are numerous errors: the paragraph in Lines 204-210 has at least 3 such errors alone ("We describe experiment setup in detail" Line 206, "aability" Line 210, missing period at the end of the paragraph).

W2 (major). Many key experimental details are not explained:
* The precise details of the policy baselines GRDPL and FIXPL are not explained. Appendix H only describes how they work on a high level.
* For the experiments in Section 3, it is not clearly described what LLM agent architecture is used. I first assumed SSA, since it was described in Section 2, but then from Section 4.2 I am led to infer that it is something simpler than SSA.
* In general, insufficient detail is given about the LLM agent architectures. Only the prompt template for SSA is provided, and no information about CoT or ReAct is provided (beyond citations).
* For the comparison of SSA and CoT/ReAct (Section 4.3), it is not explained which LLM is used (all of them? gpt-oss?).

W3. I do not find the sanity checks in Section 3.1 convincing. It is not explained why the particular metrics that are reported are necessary or sufficient to establish the claim that the simulation is sufficiently accurate. For example, it is not clear that these metrics should behave the same in human-heavy versus AI-heavy economies.

W4. I similarly do not find the capability evaluation in Section 3.2 particularly convincing. It is unclear why testing each LLM agent against two static policy agents (1 fixed, 1 greedy) is a  representative measure of capabilities. For example, are the results roughly the same if instead each LLM agent plays against 2 fixed, or 2 greedy agents? Also, while it serves as a helpful sanity check to verify whether the LLM agent outperforms the policy, a easier to interpret capability evaluation would be to benchmark its performance relative to the optimal best response.

W5. The baselines CoT/ReAct seem to be relatively weak, it would be more natural to use as baselines prior work on self-improving and reflective agents (such as the agent architectures mentioned in Appendix A).

W6 (minor). In Lines 859-863, you claim to report cost via aggregated token usage, but I don't see this information anywhere in the paper.

W7 (minor). The LLM judge does not appear to be validated, for example by comparing with human labels.

**Questions:**

Q1. There is already a rich body of prior work that studies LLM agent behavior in a simulated marketplace. Can the authors more precisely explain the novelty of their contribution relative to this prior work? (For example, the agent architecture SSA does not appear to be a novel contribution, because it appears to just be a specific instantiation of a prompt scaffold.) This is hinted at in Appendix A, however Li et al. (2024) is far from the only paper that takes a microeconomic perspective. For example, Qian et al. (2025) study bargaining and Fish et al. (2024) study pricing and auctions.

Q2. Can the authors provide the missing experimental details?

---

> ### Author Response · Authors · 2025-11-24
>
> Many thanks for your review and for highlighting some of the issues with writing + lack of experimental details. In response to W1,  W6, W7, and Q2, we updated our manuscript to improve the writing and clarify details of our experiments.
>
> > W2: Many key experimental details are not explained... policy baselines GRDPL and FIXPL... LLM agent architectures.
>
> We have updated the Appendix to further define the logic for GRDPL (budget-sorting, fixed discount) and FIXPL (static skill preference). We have also clarified that SSA, CoT, and ReAct are prompt-based architectures applied to the same underlying models (gpt-5) to control for intrinsic model capability. This ensures the comparison is strictly about the reasoning framework.
>
> > W3: I do not find the sanity checks in Section 3.1 convincing... not clear that these metrics should behave the same in human-heavy versus AI-heavy economies.
>
> To clarify, the sanity checks (e.g., the Beveridge Curve) are not intended to prove that AI economies will mimic human ones, but to validate that our simulation possesses sufficient fidelity to capture fundamental supply/demand dynamics. Without these established economic baselines, any emergent "AI-specific" behaviors (like the monopolization we observe) could be dismissed as simulation artifacts
>
> > W4: Unclear why testing each LLM agent against two static policy agents... is a representative measure.
>
> Static agents serve as "market anchors", representing the "rational" (greedy) and "bounded rational" (fixed) floors of economic behavior. Beating them demonstrates that LLM agents are not just random actors, but can formulate strategies superior to heuristic algorithms. While we agree that a theoretical "optimal best response" would be a good substitute, calculating this in a partially observable stochastic game with heterogeneous agents is computationally intractable, making these baselines the standard approach in agent-based computational economics
>
> > W5: The baselines CoT/ReAct seem to be relatively weak, it would be more natural to use as baselines prior work on self-improving and reflective agents (such as the agent architectures mentioned in Appendix A).
>
> We recognize the potential value of comparing against more complex agent architectures such as Reflexion. However, the main goal of our paper was to illustrate the potential labor market dynamics of AI agents, and we chose CoT and ReAct as baselines because they represent strong, widely-adopted, and simple reasoning and action-loop scaffolds that control for the underlying LLM's intrinsic capability. In addition, integrating the more complex agent architectures into our framework presented significant architectural and computational challenges. Furthermore, the more complex architecture requiring intensive self-refinement loops incurred significantly higher token costs than the ReAct and CoT baselines, and given that our framework evaluates agents over 100+ turns there were resource constraints. Nonetheless, we agree that comparing SSA against more contemporary agent architecture would be a valuable direction for future research.
>
> > Q1. Can the authors more precisely explain the novelty of their contribution relative to this prior work? (The agent architecture SSA does not appear to be a novel contribution...)
>
> Our novelty is fundamentally about the framework and mechanism design of the labor market, not the agent architecture. While previous work study pricing or bargaining, none integrate the critical labor market trade-offs of Adverse Selection (hidden skill) and Moral Hazard (unseen effort) with the agent's meta-action of self-improvement v.s. maximising income. Our contribution is creating a context where agents reason about economic value (training ROI) rather than just task performance, and the SSA architecture is an illustrative example of the high level capabilities that agents would need to possess to succeed in this market.

---

### Official Review · Reviewer_LRqr · 2025-11-01

**Soundness:** 2
**Presentation:** 3
**Contribution:** 2
**Rating:** 4
**Confidence:** 3

**Summary:**

This paper develops a formal framework and a simulation platform — AI Work — to study labour-market dynamics when autonomous AI agents (LLM-based) compete for jobs, invest in skills, and adapt strategies over repeated rounds. The authors model the market as a competitive skill-based stochastic game, implement a gig-economy style simulator (with reputation, bidding, training, capacity constraints, etc.) and experiments on large-scale fixed-policy simulations to study emergent macroeconomic patterns (e.g., Beveridge-like curves, concentration), large-scale fixed-policy simulations to study emergent macroeconomic patterns (e.g., Beveridge-like curves, concentration).

**Strengths:**

1. The question of how LLM agents behave in economic settings is societally and scientifically relevant; the paper tackles a forward-looking problem with potential policy and ML implications.
2. The market formalization (Competitive Skill-Based Stochastic Game) and the engineering of AI Work (price–reputation scoring, stochastic reranking, capacity constraints, skill/reputation dynamics) are well specified and grounded in economic primitives (Cobb–Douglas/CES scoring, Beta reputation aggregation).

**Weaknesses:**

1. Simplified proxy tasks and utility model. The simulated tasks are proxy tasks with stochastic scoring; how sensitive are conclusions (e.g., monopolization, wage deflation) to the choice of task scoring function $\gamma$(·), client preference generation P(·), or to the Cobb–Douglas aggregator/parameterization (wq, wp)? The manuscript argues qualitative alignment with known macro facts, but it lacks sensitivity analyses showing results are robust across realistic alternative parameterizations.
2. The environment hides latent skills and permits only price & reputation observations — a reasonable simplification — but many real markets expose additional signals (samples, portfolios, verification). It is unclear how adding richer observability or dispute/verification mechanics would change results. The limitations section mentions simplifications; more systematic exploration or discussion is needed.
3. Ablation interpretability. The paper reports that metacognition is the dominant capability, planning less so. This is plausible, but further analysis—e.g., causal ablation, counterfactual prompting, or controlled micro-tasks isolating each capability—would strengthen the claim and assure it’s not an artifact of prompt phrasing.

**Questions:**

1. Would the key macro and micro claims hold under variations of: (a) reputation update parameters (λ, W, H), (b) score aggregator weights (wq, wp) and CES parameter ρ, and (c) job/task diversity and concurrent capacity ν. Report whether monopolization and wage deflation persist qualitatively.
2. How real-world platform features (verification, sample portfolios, dispute resolution) might change results and list concrete platform design levers that could mitigate harms (e.g., reputation caps, randomized matching, diversity quotas)

**Details Of Ethics Concerns:**

The paper briefly mentions potential harms (monopolization, displacement), but policy implications and mitigation strategies (platform design interventions, regulation) are lightly sketched. Given the real-world stakes, the discussion could be expanded with more specific mechanism design recommendations and ethical considerations (e.g., human worker protection, transparency).

---

> ### Author Response · Authors · 2025-11-24
>
> Many thanks for your review and feedback! In response to some of the comments and questions:
>
> > Simplified proxy tasks and utility model... it lacks sensitivity analyses showing results are robust across realistic alternative parameterizations.
>
> Our simulation framework includes both proxy and real tasks. However, during exploratory analysis we noted that proxy tasks allowed for a more tractable way to simulate the economy, as our goal was to isolate the economic reasoning of agents (bidding vs. training decisions) from the noise of task-specific difficulty. While we agree that full sensitivity analyses would be valuable, our current results demonstrate that the mechanisms of the market (e.g., the pressure to underbid in open auctions) function correctly according to economic theory. Extensive analyses would characterize specific simulation instances, whereas our contribution is the framework that allows such forces to be modeled in a contained, traceable manner
>
> > The environment hides latent skills... It is unclear how adding richer observability... would change results.
>
> Hidden latent skills are strictly necessary to model adverse selection, which is one of the core economic forces we aim to study. If skills were fully observable, the market would reduce to a simple optimization problem rather than a game of information asymmetry and reputation building.
>
> > Ablation interpretability... causal ablation... would strengthen the claim.
>
> We agree that causal analysis is the gold standard. However, the prompt ablation study (comparing SSA against CoT/ReAct while controlling for the underlying model) is a form of causal intervention on the agent's reasoning process. The correlation between the presence of metacognitive reasoning traces and economic performance ($r=0.744$) provides evidence that these capabilities are drivers of success in a competitive market
>
> > Would the key macro and micro claims hold under variations of... Report whether monopolization and wage deflation persist qualitatively.
>
> We agree that a comprehensive parameter sweep is vital for characterizing a fully deployed system; however, this level of sensitivity analysis falls outside the scope of this foundational framework paper, which is resource- and timeline-constrained. Our goal was to establish that the market mechanisms possess sufficient stability to exhibit the core phenomena. In our expeirments we explored some parts of this space, e.g. varying concurrent job capacity $\nu$ demonstrates the acceleration of monopolization (Figure 3C), and varying the score aggregator weights ($w_q, w_p$) via the price sensitivity factor confirms the stability of wage deflation under certain market incentives (Figure 6A). These initial results consistently show that monopolization and wage deflation persist qualitatively, confirming the framework's stability under parameter deviations.
>
> > How real-world platform features (verification, sample portfolios, dispute resolution) might change results and list concrete platform design levers that could mitigate harms (e.g., reputation caps, randomized matching, diversity quotas)
>
> We appreciate the suggestions re real-world platform features that would add to our contribution on economic mechanism design. We added a section in discussion detailing how explicit verification mechanisms would theoretically substitute reputation and accelerate price competition. In our paper, we discussed some existing mitigation levers, including (i) Sealed Bidding (to counter wage deflation and encourage training), (ii) Capacity Constraints $\nu$ (to curb monopolization), and (iii) Diversity-Aware Matching or specialization incentives (to mitigate inequality). Other suggestions, such as including sample portfolios or dispute resolution mechanics, are interesting features we would look to implement in future research

---

### Author Response · Authors · 2025-11-24
**Clarifying our paper as a Framework to study Economic Forces in AI Labor Markets**

We thank the reviewers for their comprehensive and insightful feedback. We acknowledge the valid concerns regarding the paper's initial presentation and the desire for more extensive parameter sensitivity analysis. However, in reading the comments, we believe the central goal of our work was misunderstood.

This submission is not intended to be a large-scale macroeconomic simulation, nor is it a contribution to the existing multi-agent literature on task-specific optimization. The aim of this paper was to be foundational first and foremost, where we introduce the first framework capable of capturing the fundamental microeconomic forces of adverse selection, moral hazard, and reputation dynamics within an LLM-agent labor market.

As such, the purpose of our simulation is to serve as a tractable testbed to analyze how these specific economic pressures drive the spontaneous development of strategic LLM capabilities (Metacognition, Competitive Awareness) and reveal platform-design-dependent outcomes (e.g., systemic wage deflation under open bidding). We believe that establishing these mechanisms in our framework would be useful for future research into AI agents within the economy, and attempting large-scale, high-fidelity replication of current human labor markets is not the main goal of this paper nor is it possible given the number of potential levers to simulate, as highlighted by multiple reviewers.

In response to the reviewers and to reflect the above, we made several changes to our manuscript:
1. Narrative clarity: We rewrote the introduction, and added a new section to explicitly define the economic forces our framework isolates. Section 2 now explicitly details the implementation of Adverse Selection and Moral Hazard. We hope to clarify that our paper's value lies in uniting economic mechanism design with strategic LLM behavior, and distinguish our work from general multi-agent economic simulations
2. Clarified Experimental Details: We updated Appendices G and H (defining GRDPL/FIXPL algorithms) and added Appendix L (exact prompt structures for CoT/ReAct) to ensure reproducibility.
3. Addressed Validity: We added details to Appendix J regarding our human-in-the-loop validation process for the SSA scoring rubric.

---

### Meta-Review · Area_Chair_Lz4m · 2026-01-04

**Summary:**

1. The proposed gig economy is a simplified model of practice. The robustness has not been addressed.
2. The novelty compared to the prior model has not been clearly clarified, e.g., the necessity of introducing SSA scoring

**Reviewer Concerns:**

The rebuttal partially addresses the concern about the simplified proxy tasks and utility model. But there is still concern about its robustness. It will be helpful to provide robustness analysis for e.g., the choice of task scoring function or client preference generation

**Reviewer Scores:**

No change

---

### Decision · Program_Chairs · 2026-01-26

Reject